# Selective Classification for Deep Neural Networks

**Yonatan Geifman**
Computer Science Department
Technion – Israel Institute of Technology
yonatan.g@cs.technion.ac.il

**Ran El-Yaniv**
Computer Science Department
Technion – Israel Institute of Technology
rani@cs.technion.ac.il

## Abstract

Selective classification techniques (also known as reject option) have not yet been considered in the context of deep neural networks (DNNs). These techniques can potentially significantly improve DNNs prediction performance by trading-off coverage. In this paper we propose a method to construct a selective classifier given a trained neural network. Our method allows a user to set a desired risk level. At test time, the classifier rejects instances as needed, to grant the desired risk (with high probability). Empirical results over CIFAR and ImageNet convincingly demonstrate the viability of our method, which opens up possibilities to operate DNNs in mission-critical applications. For example, using our method an unprecedented 2% error in top-5 ImageNet classification can be guaranteed with probability 99.9%, and almost 60% test coverage.

## 1 Introduction

While self-awareness remains an illusive, hard to define concept, a rudimentary kind of self-awareness, which is much easier to grasp, is the ability to know what you don't know, which can make you smarter. The subfield dealing with such capabilities in machine learning is called *selective prediction* (also known as prediction with a *reject option*), which has been around for 60 years [1, 5]. The main motivation for selective prediction is to reduce the error rate by abstaining from prediction when in doubt, while keeping coverage as high as possible. An ultimate manifestation of selective prediction is a classifier equipped with a "dial" that allows for precise control of the desired true error rate (which should be guaranteed with high probability), while keeping the coverage of the classifier as high as possible.

Many present and future tasks performed by (deep) predictive models can be dramatically enhanced by high quality selective prediction. Consider, for example, autonomous driving. Since we cannot rely on the advent of "singularity", where AI is superhuman, we must manage with standard machine learning, which sometimes errs. But what if our deep autonomous driving network were capable of knowing that it doesn't know how to respond in a certain situation, disengaging itself in advance and alerting the human driver (hopefully not sleeping at that time) to take over? There are plenty of other mission-critical applications that would likewise greatly benefit from effective selective prediction.

The literature on the reject option is quite extensive and mainly discusses rejection mechanisms for various hypothesis classes and learning algorithms, such as SVM, boosting, and nearest-neighbors [8, 13, 3]. The reject option has rarely been discussed in the context of neural networks (NNs), and so far has not been considered for deep NNs (DNNs). Existing NN works consider a cost-based rejection model [2, 4], whereby the costs of misclassification and abstaining must be specified, and a rejection mechanism is optimized for these costs. The proposed mechanism for classification is based on applying a carefully selected threshold on the maximal neuronal response of the softmax layer. We that call this mechanism *softmax response* (SR). The cost model can be very useful when we can quantify the involved costs, but in many applications of interest meaningful costs are hard to reason. (Imagine trying to set up appropriate rejection/misclassification costs for disengaging an autopilot

driving system.) Here we consider the alternative risk-coverage view for selective classification discussed in [5].

Ensemble techniques have been considered for selective (and confidence-rated) prediction, where rejection mechanisms are typically based on the ensemble statistics [18, 7]. However, such techniques are presently hard to realize in the context of DNNs, for which it could be very costly to train sufficiently many ensemble members. Recently, Gal and Ghahramani [9] proposed an ensemble-like method for measuring uncertainty in DNNs, which bypasses the need to train several ensemble members. Their method works via sampling multiple dropout applications of the forward pass to perturb the network prediction randomly. While this Monte-Carlo dropout (MC-dropout) technique was not mentioned in the context of selective prediction, it can be directly applied as a viable selective prediction method using a threshold, as we discuss here.

In this paper we consider classification tasks, and our goal is to learn a selective classifier $(f, g)$, where $f$ is a standard classifier and $g$ is a rejection function. The selective classifier has to allow full *guaranteed* control over the true risk. The ideal method should be able to classify samples in production with any desired level of risk with the *optimal* coverage rate. It is reasonable to assume that this optimal performance can only be obtained if the pair $(f, g)$ is trained together. As a first step, however, we consider a simpler setting where a (deep) neural classifier $f$ is already given, and our goal is to learn a rejection function $g$ that will guarantee with high probability a desired error rate. To this end, we consider the above two known techniques for rejection (SR and MC-dropout), and devise a learning method that chooses an appropriate threshold that ensures the desired risk. For a given classifier $f$, confidence level $\delta$, and desired risk $r^*$, our method outputs a selective classifier $(f, g)$ whose test error will be no larger than $r^*$ with probability of at least $1 - \delta$.

Using the well-known VGG-16 architecture, we apply our method on CIFAR-10, CIFAR-100 and ImageNet (on ImageNet we also apply the RESNET-50 architecture). We show that both SR and dropout lead to extremely effective selective classification. On both the CIFAR datasets, these two mechanisms achieve nearly identical results. However, on ImageNet, the simpler SR mechanism is significantly superior. More importantly, we show that almost any desirable risk level can be guaranteed with a surprisingly high coverage. For example, an unprecedented 2% error in top-5 ImageNet classification can be guaranteed with probability 99.9%, and almost 60% test coverage.

## 2 Problem Setting

We consider a standard multi-class classification problem. Let $\mathcal{X}$ be some feature space (e.g., raw image data) and $\mathcal{Y}$, a finite label set, $\mathcal{Y} = \{1, 2, 3, \ldots, k\}$, representing $k$ classes. Let $P(X, Y)$ be a distribution over $\mathcal{X} \times \mathcal{Y}$. A classifier $f$ is a function $f : \mathcal{X} \to \mathcal{Y}$, and the *true risk* of $f$ w.r.t. $P$ is $R(f|P) \triangleq E_{P(X,Y)}[\ell(f(x), y)]$, where $\ell : Y \times Y \to \mathbb{R}^+$ is a given loss function, for example the 0/1 error. Given a labeled set $S_m = \{(x_i, y_i)\}_{i=1}^m \subseteq (\mathcal{X} \times \mathcal{Y})$ sampled i.i.d. from $P(X, Y)$, the *empirical risk* of the classifier $f$ is $\hat{r}(f|S_m) \triangleq \frac{1}{m} \sum_{i=1}^m \ell(f(x_i), y_i)$.

A *selective classifier* [5] is a pair $(f, g)$, where $f$ is a classifier, and $g : \mathcal{X} \to \{0, 1\}$ is a *selection function*, which serves as a binary qualifier for $f$ as follows,

$$(f, g)(x) \triangleq \begin{cases} f(x), & \text{if } g(x) = 1; \\ \text{don't know}, & \text{if } g(x) = 0. \end{cases}$$

Thus, the selective classifier abstains from prediction at a point $x$ iff $g(x) = 0$. The performance of a selective classifier is quantified using *coverage* and *risk*. Fixing $P$, coverage, defined to be $\phi(f, g) \triangleq E_P[g(x)]$, is the probability mass of the non-rejected region in $\mathcal{X}$. The selective risk of $(f, g)$ is

$$R(f, g) \triangleq \frac{E_P[\ell(f(x), y)g(x)]}{\phi(f, g)}. \tag{1}$$

Clearly, the risk of a selective classifier can be traded-off for coverage. The entire performance profile of such a classifier can be specified by its *risk-coverage* curve, defined to be risk as a function of coverage [5].

Consider the following problem. We are given a classifier $f$, a training sample $S_m$, a *confidence* parameter $\delta > 0$, and a desired *risk target* $r^* > 0$. Our goal is to use $S_m$ to create a selection function

$g$ such that the selective risk of $(f, g)$ satisfies

$$\mathbf{Pr}_{S_m} \{R(f, g) > r^*\} < \delta, \tag{2}$$

where the probability is over training samples, $S_m$, sampled i.i.d. from the unknown underlying distribution $P$. Among all classifiers satisfying (2), the best ones are those that maximize the coverage. For a fixed $f$, and a given class $\mathcal{G}$ (which will be discussed below), in this paper our goal is to select $g \in \mathcal{G}$ such that the selective risk $R(f, g)$ satisfies (2) while the coverage $\Phi(f, g)$. is maximized.

## 3   Selection with Guaranteed Risk Control

In this section, we present a general technique for constructing a selection function with guaranteed performance, based on a given classifier $f$, and a confidence-rate function $\kappa_f : \mathcal{X} \to \mathbb{R}^+$ for $f$. We do not assume anything on $\kappa_f$, and the *interpretation* is that $\kappa$ can rank in the sense that if $\kappa_f(x_1) \geq \kappa_f(x_2)$, for $x_1, x_2 \in \mathcal{X}$, the confidence function $\kappa_f$ indicates that the confidence in the prediction $f(x_2)$ is not higher than the confidence in the prediction $f(x_1)$. In this section we are not concerned with the question of what is a good $\kappa_f$ (which is discussed in Section 4); our goal is to generate a selection function $g$, with guaranteed performance for a given $\kappa_f$.

For the reminder of this paper, the loss function $\ell$ is taken to be the standard 0/1 loss function (unless explicitly mentioned otherwise). Let $S_m = \{(x_i, y_i)\}_{i=1}^{m} \subseteq (\mathcal{X} \times \mathcal{Y})^m$ be a training set, assumed to be sampled i.i.d. from an unknown distribution $P(X, Y)$. Given also are a *confidence* parameter $\delta > 0$, and a desired *risk target* $r^* > 0$. Based on $S_m$, our goal is to learn a selection function $g$ such that the selective risk of the classifier $(f, g)$ satisfies (2).

For $\theta > 0$, we define the selection function $g_\theta : \mathcal{X} \to \{0, 1\}$ as

$$g_\theta(x) = g_\theta(x|\kappa_f) \triangleq \left\{ \begin{array}{ll} 1, & \text{if } \kappa_f(x) \geq \theta; \\ 0, & \text{otherwise.} \end{array} \right. \tag{3}$$

For any selective classifier $(f, g)$, we define its *empirical selective risk* with respect to the labeled sample $S_m$,

$$\hat{r}(f, g|S_m) \triangleq \frac{\frac{1}{m} \sum_{i=1}^{m} \ell(f(x_i), y_i) g(x_i)}{\hat{\phi}(f, g|S_m)},$$

where $\hat{\phi}$ is the *empirical coverage*, $\hat{\phi}(f, g|S_m) \triangleq \frac{1}{m} \sum_{i=1}^{m} g(x_i)$. For any selection function $g$, denote by $g(S_m)$ the $g$-projection of $S_m$, $g(S_m) \triangleq \{(x, y) \in S_m \ : \ g(x) = 1\}$.

The *selection with guaranteed risk* (SGR) learning algorithm appears in Algorithm 1. The algorithm receives as input a classifier $f$, a confidence-rate function $\kappa_f$, a confidence parameter $\delta > 0$, a target risk $r^{*1}$, and a training set $S_m$. The algorithm performs a binary search to find the optimal bound guaranteeing the required risk with sufficient confidence. The SGR algorithm outputs a selective classifier $(f, g)$ and a risk bound $b^*$. In the rest of this section we analyze the SGR algorithm. We make use of the following lemma, which gives the tightest possible numerical generalization bound for a single classifier, based on a test over a labeled sample.

**Lemma 3.1 (Gascuel and Caraux, 1992, [10])** *Let $P$ be any distribution and consider a classifier $f$ whose true error w.r.t. $P$ is $R(f|P)$. Let $0 < \delta < 1$ be given and let $\hat{r}(f|S_m)$ be the empirical error of $f$ w.r.t. to the labeled set $S_m$, sampled i.i.d. from $P$. Let $B^*(\hat{r}_i, \delta, S_m)$ be the solution $b$ of the following equation,*

$$\sum_{j=0}^{m \cdot \hat{r}(f|S_m)} \binom{m}{j} b^j (1-b)^{m-j} = \delta. \tag{4}$$

*Then, $\mathbf{Pr}_{S_m} \{R(f|P) > B^*(\hat{r}_i, \delta, S_m)\} < \delta$.*

We emphasize that the numerical bound of Lemma 3.1 is the tightest possible in this setting. As discussed in [10], the analytic bounds derived using, e.g., Hoeffding inequality (or other concentration inequalities), approximate this numerical bound and incur some slack.

**Algorithm 1** *Selection with Guaranteed Risk* (SGR)
---
1: SGR($f, \kappa_f, \delta, r^*, S_m$)
2: Sort $S_m$ according to $\kappa_f(x_i)$, $x_i \in S_m$ (and now assume w.l.o.g. that indices reflect this ordering).
3: $z_{\min} = 1$; $z_{\max} = m$
4: **for** $i = 1$ **to** $k \triangleq \lceil \log_2 m \rceil$ **do**
5:      $z = \lceil (z_{\min} + z_{\max})/2 \rceil$
6:      $\theta = \kappa_f(x_z)$
7:      $g_i = g_\theta$ {(see (3))}
8:      $\hat{r}_i = \hat{r}(f, g_i | S_m)$
9:      $b_i^* = B^*(\hat{r}_i, \delta/\lceil \log_2 m \rceil, g_i(S_m))$ {see Lemma 3.1 }
10:      **if** $b_i^* < r^*$ **then**
11:         $z_{\max} = z$
12:      **else**
13:         $z_{\min} = z$
14:      **end if**
15: **end for**
16: Output- $(f, g_k)$ and the bound $b_k^*$.
---

For any selection function, $g$, let $P_g(X, Y)$ be the projection of $P$ over $g$; that is, $P_g(X, Y) \triangleq P(X, Y | g(X) = 1)$. The following theorem is a uniform convergence result for the SGR procedure.

**Theorem 3.2 (SGR)** *Let $S_m$ be a given labeled set, sampled i.i.d. from $P$, and consider an application of the SGR procedure. For $k \triangleq \lceil \log_2 m \rceil$, let $(f, g_i)$ and $b_i^*$, $i = 1, \ldots, k$, be the selective classifier and bound computed by SGR in its ith iterations. Then,*

$$\mathbf{Pr}_{S_m} \{\exists i : R(f | P_{g_i}) > B^*(\hat{r}_i, \delta/k, g_i(S_m))\} < \delta.$$

**Proof Sketch:** For any $i = 1, \ldots, k$, let $m_i = |g_i(S_m)|$ be the random variable giving the size of accepted examples from $S_m$ on the $i$th iteration of SGR. For any fixed value of $0 \leq m_i \leq m$, by Lemma 3.1, applied with the projected distribution $P_{g_i}(X, Y)$, and a sample $S_{m_i}$, consisting of $m_i$ examples drawn from the product distribution $(P_{g_i})^{m_i}$,

$$\mathbf{Pr}_{S_{m_i} \sim (P_{g_i})^{m_i}} \{R(f | P_{g_i}) > B^*(\hat{r}_i, \delta/k, g_i(S_m))\} < \delta/k. \tag{5}$$

The sampling distribution of $m_i$ labeled examples in SGR is determined by the following process: sample a set $S_m$ of $m$ examples from the product distribution $P^m$ and then use $g_i$ to filter $S_m$, resulting in a (randon) number $m_i$ of examples. Therefore, the left-hand side of (5) equals

$$\mathbf{Pr}_{S_m \sim P^m} \{R(f | P_{g_i}) > B^*(\hat{r}_i, \delta/k, g_i(S_m)) \mid g_i(S_m) = m_i\}.$$

Clearly,

$$R(f | P_{g_i}) = E_{P_{g_i}}[\ell(f(x), y)] = \frac{E_P[\ell(f(x), y) g(x)]}{\phi(f, g)} = R(f, g_i).$$

Therefore,

$$\mathbf{Pr}_{S_m} \{R(f, g_i) > B^*(\hat{r}_i, \delta/k, g_i(S_m))\}$$
$$= \sum_{n=0}^{m} \mathbf{Pr}_{S_m} \{R(f, g_i) > B^*(\hat{r}_i, \delta/k, g_i(S_m)) \mid g_i(S_m) = n\} \cdot \mathbf{Pr}\{g_i(S_m) = n\}$$
$$\leq \frac{\delta}{k} \sum_{n=0}^{m} \mathbf{Pr}\{g_i(S_m) = n\} = \frac{\delta}{k}.$$

An application of the union bound completes the proof. $\qquad\square$

## 4 Confidence-Rate Functions for Neural Networks

Consider a classifier $f$, assumed to be trained for some unknown distribution $P$. In this section we consider two confidence-rate functions, $\kappa_f$, based on previous work [9, 2]. We note that an ideal

confidence-rate function $\kappa_f(x)$ for $f$, should reflect true loss monotonicity. Given $(x_1, y_1) \sim P$ and $(x_2, y_2) \sim P$, we would like the following to hold: $\kappa_f(x_1) \leq \kappa_f(x_2)$ if and only if $\ell(f(x_1), y_1) \geq \ell(f(x_2), y_2)$. Obviously, one cannot expect to have an ideal $\kappa_f$. Given a confidence-rate functions $\kappa_f$, a useful way to analyze its effectiveness is to draw the risk-coverage curve of its induced rejection function, $g_\theta(x|\kappa_f)$, as defined in (3). This risk-coverage curve shows the relationship between $\theta$ and $R(f, g_\theta)$. For example, see Figure 2(a) where a two (nearly identical) risk-coverage curves are plotted. While the confidence-rate functions we consider are not ideal, they will be shown empirically to be extremely effective. [2]

The first confidence-rate function we consider has been around in the NN folklore for years, and is explicitly mentioned by [2, 4] in the context of reject option. This function works as follows: given any neural network classifier $f(x)$ where the last layer is a softmax, we denote by $f(x|j)$ the soft response output for the $j$th class. The confidence-rate function is defined as $\kappa \triangleq \max_{j \in \mathcal{Y}}(f(x|j))$. We call this function *softmax response* (SR).

Softmax responses are often treated as probabilities (responses are positive and sum to 1), but some authors criticize this approach [9]. Noting that, for our purposes, the ideal confidence-rate function should only provide coherent *ranking* rather than absolute probability values, softmax responses are potentially good candidates for *relative* confidence rates.

We are not familiar with a rigorous explanation for SR, but it can be intuitively motivated by observing neuron activations. For example, Figure 1 depicts average response values of every neuron in the second-to-last layer for true positives and false positives for the class '8' in the MNIST dataset (and qualitatively similar behavior occurs in all MNIST classes). The x-axis corresponds to neuron indices in that layer (1-128); and the y-axis shows the average responses, where green squares are averages of true positives, boldface squares highlight strong responses, and red circles correspond to the average response of false positives. It is evident that the true positive activation response in the active neurons is much higher than the false positive, which is expected to be reflected in the final softmax layer response. Moreover, it can be seen that the large activation values are spread over many neurons, indicating that the confidence signal arises due to numerous patterns detected by neurons in this layer. Qualitatively similar behavior can be observed in deeper layers.

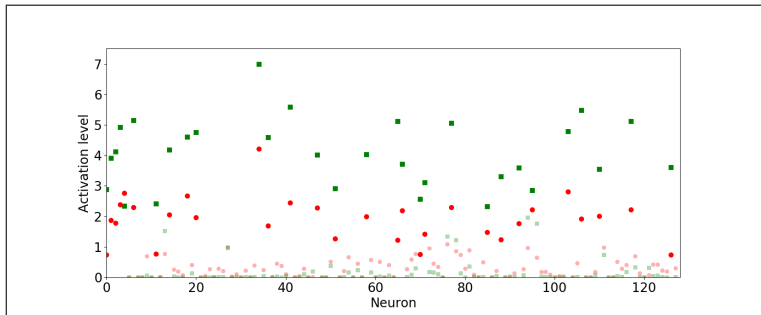

Figure 1: Average response values of neuron activations for class "8" on the MNIST dataset; green squares, true positives, red circles, false negatives

The MC-dropout technique we consider was recently proposed to quantify uncertainty in neural networks [9]. To estimate uncertainty for a given instance $x$, we run a number of feed-forward iterations over $x$, each applied with dropout in the last fully connected layer. Uncertainty is taken as the variance in the responses of the neuron corresponding to the most probable class. We consider minus uncertainty as the MC-dropout confidence rate.

## 5    Empirical Results

In Section 4 we introduced the SR and MC-dropout confidence-rate function, defined for a given model $f$. We trained VGG models [17] for CIFAR-10, CIFAR-100 and ImageNet. For each of these models $f$, we considered both the SR and MC-dropout confidence-rate functions, $\kappa_f$, and the induced

rejection function, $g_\theta(x|\kappa_f)$. In Figure 2 we present the risk-coverage curves obtained for each of the three datasets. These curves were obtained by computing a validation risk and coverage for many $\theta$ values. It is evident that the risk-coverage profile for SR and MC-dropout is nearly identical for both the CIFAR datasets. For the ImageNet set we plot the curves corresponding to top-1 (dashed curves) and top-5 tasks (solid curves). On this dataset, we see that SR is significantly better than MC-dropout on both tasks. For example, in the top-1 task and 60% coverage, the SR rejection has 10% error while MC-dropout rejection incurs more than 20% error. But most importantly, these risk-coverage curves show that selective classification can potentially be used to dramatically reduce the error in the three datasets. Due to the relative advantage of SR, in the rest of our experiments we only focus on the SR rating.

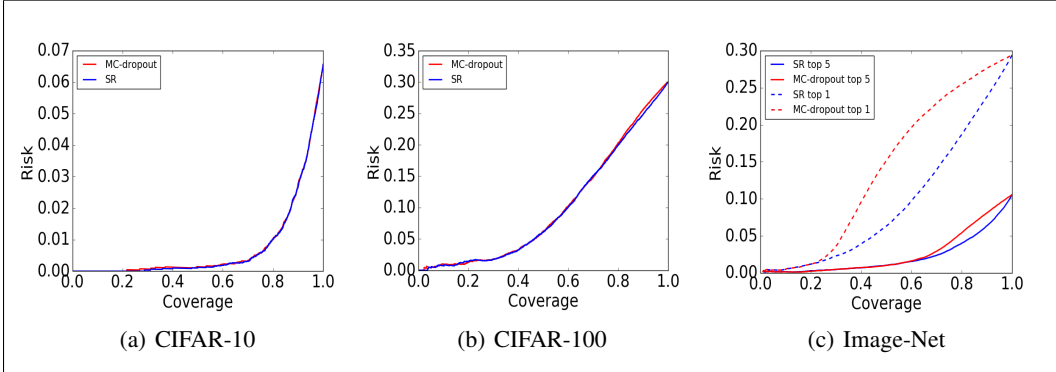

|  | (a) CIFAR-10 | (b) CIFAR-100 | (c) Image-Net |

Figure 2: Risk coverage curves for (a) cifar-10, (b) cifar-100 and (c) image-net (top-1 task: dashed curves; top-5 task: solid crves), SR method in blue and MC-dropout in red.

We now report on experiments with our SGR routine, and apply it on each of the datasets to construct high probability risk-controlled selective classifiers for the three datasets.

Table 1: Risk control results for CIFAR-10 for $\delta = 0.001$

| Desired risk ($r^*$) | Train risk | Train coverage | Test risk | Test coverage | Risk bound ($b^*$) |
|---|---|---|---|---|---|
| 0.01 | 0.0079 | 0.7822 | 0.0092 | 0.7856 | 0.0099 |
| 0.02 | 0.0160 | 0.8482 | 0.0149 | 0.8466 | 0.0199 |
| 0.03 | 0.0260 | 0.8988 | 0.0261 | 0.8966 | 0.0298 |
| 0.04 | 0.0362 | 0.9348 | 0.0380 | 0.9318 | 0.0399 |
| 0.05 | 0.0454 | 0.9610 | 0.0486 | 0.9596 | 0.0491 |
| 0.06 | 0.0526 | 0.9778 | 0.0572 | 0.9784 | 0.0600 |

## 5.1 Selective Guaranteed Risk for CIFAR-10

We now consider CIFAR-10; see [14] for details. We used the VGG-16 architecture [17] and adapted it to the CIFAR-10 dataset by adding massive dropout, exactly as described in [15]. We used data augmentation containing horizontal flips, vertical and horizontal shifts, and rotations, and trained using SGD with momentum of 0.9, initial learning rate of 0.1, and weight decay of 0.0005. We multiplicatively dropped the learning rate by 0.5 every 25 epochs, and trained for 250 epochs. With this setting we reached validation accuracy of 93.54, and used the resulting network $f_{10}$ as the basis for our selective classifier.

We applied the SGR algorithm on $f_{10}$ with the SR confidence-rating function, where the training set for SGR, $S_m$, was taken as half of the standard CIFAR-10 validation set that was randomly split to two equal parts. The other half, which was not consumed by SGR for training, was reserved for testing the resulting bounds. Thus, this training and test sets where each of approximately 5000 samples. We applied the SGR routine with several desired risk values, $r^*$, and obtained, for each such $r^*$, corresponding selective classifier and risk bound $b^*$. All our applications of the SGR routine

(for this dataset and the rest) where with a particularly small confidence level $\delta = 0.001$.[3] We then applied these selective classifiers on the reserved test set, and computed, for each selective classifier, *test risk* and *test coverage*. The results are summarized in Table 1, where we also include *train risk* and *train coverage* that were computed, for each selective classifier, over the training set.

Observing the results in Table 1, we see that the risk bound, $b^*$, is always very close to the target risk, $r^*$. Moreover, the test risk is always bounded above by the bound $b^*$, as required. We compared this result to a basic baseline in which the threshold is defined to be the value that maximizes coverage while keeping train error smaller then $r^*$. For this simple baseline we found that in over 50% of the cases (1000 random train/test splits), the bound $r^*$ was violated over the test set, with a mean violation of 18% relative to the requested $r^*$. Finally, we see that it is possible to guarantee with this method amazingly small 1% error while covering more than 78% of the domain.

## 5.2 Selective Guaranteed Risk for CIFAR-100

Using the same VGG architechture (now adapted to 100 classes) we trained a model for CIFAR-100 while applying the same data augmentation routine as in the CIFAR-10 experiment. Following precisly the same experimental design as in the CFAR-10 case, we obtained the results of Table 2

Table 2: Risk control results for CIFAR-100 for $\delta = 0.001$

| Desired risk ($r^*$) | Train risk | Train coverage | Test risk | Test coverage | Risk bound ($b^*$) |
|---|---|---|---|---|---|
| 0.02 | 0.0119 | 0.2010 | 0.0187 | 0.2134 | 0.0197 |
| 0.05 | 0.0425 | 0.4286 | 0.0413 | 0.4450 | 0.0499 |
| 0.10 | 0.0927 | 0.5736 | 0.0938 | 0.5952 | 0.0998 |
| 0.15 | 0.1363 | 0.6546 | 0.1327 | 0.6752 | 0.1498 |
| 0.20 | 0.1872 | 0.7650 | 0.1810 | 0.7778 | 0.1999 |
| 0.25 | 0.2380 | 0.8716 | 0.2395 | 0.8826 | 0.2499 |

Here again, SGR generated tight bounds, very close to the desired target risk, and the bounds were never violated by the true risk. Also, we see again that it is possible to dramatically reduce the risk with only moderate compromise of the coverage. While the architecture we used is not state-of-the art, with a coverage of 67%, we easily surpassed the best known result for CIFAR-100, which currently stands on 18.85% using the wide residual network architecture [19]. It is very likely that by using ourselves the wide residual network architecture we could obtain significantly better results.

## 5.3 Selective Guaranteed Risk for ImageNet

We used an already trained Image-Net VGG-16 model based on ILSVRC2014 [16]. We repeated the same experimental design but now the sizes of the training and test set were approximately 25,000. The SGR results for both the top-1 and top-5 classification tasks are summarized in Tables 3 and 4, respectively. We also implemented the RESNET-50 architecture [12] in order to see if qualitatively similar results can be obtained with a different architecture. The RESNET-50 results for ImageNet top-1 and top-5 classification tasks are summarized in Tables 5 and 6, respectively.

Table 3: SGR results for Image-Net dataset using VGG-16 top-1 for $\delta = 0.001$

| Desired risk ($r^*$) | Train risk | Train coverage | Test risk | Test coverage | Risk bound($b^*$) |
|---|---|---|---|---|---|
| 0.02 | 0.0161 | 0.2355 | 0.0131 | 0.2322 | 0.0200 |
| 0.05 | 0.0462 | 0.4292 | 0.0446 | 0.4276 | 0.0500 |
| 0.10 | 0.0964 | 0.5968 | 0.0948 | 0.5951 | 0.1000 |
| 0.15 | 0.1466 | 0.7164 | 0.1467 | 0.7138 | 0.1500 |
| 0.20 | 0.1937 | 0.8131 | 0.1949 | 0.8154 | 0.2000 |
| 0.25 | 0.2441 | 0.9117 | 0.2445 | 0.9120 | 0.2500 |

Table 4: SGR results for Image-Net dataset using VGG-16 top-5 for $\delta = 0.001$

| Desired risk ($r^*$) | Train risk | Train coverage | Test risk | Test coverage | Risk bound($b^*$) |
|---|---|---|---|---|---|
| 0.01 | 0.0080 | 0.3391 | 0.0078 | 0.3341 | 0.0100 |
| 0.02 | 0.0181 | 0.5360 | 0.0179 | 0.5351 | 0.0200 |
| 0.03 | 0.0281 | 0.6768 | 0.0290 | 0.6735 | 0.0300 |
| 0.04 | 0.0381 | 0.7610 | 0.0379 | 0.7586 | 0.0400 |
| 0.05 | 0.0481 | 0.8263 | 0.0496 | 0.8262 | 0.0500 |
| 0.06 | 0.0563 | 0.8654 | 0.0577 | 0.8668 | 0.0600 |
| 0.07 | 0.0663 | 0.9093 | 0.0694 | 0.9114 | 0.0700 |

Table 5: SGR results for Image-Net dataset using RESNET50 top-1 for $\delta = 0.001$

| Desired risk ($r^*$) | Train risk | Train coverage | Test risk | Test coverage | Risk bound ($b^*$) |
|---|---|---|---|---|---|
| 0.02 | 0.0161 | 0.2613 | 0.0164 | 0.2585 | 0.0199 |
| 0.05 | 0.0462 | 0.4906 | 0.0474 | 0.4878 | 0.0500 |
| 0.10 | 0.0965 | 0.6544 | 0.0988 | 0.6502 | 0.1000 |
| 0.15 | 0.1466 | 0.7711 | 0.1475 | 0.7676 | 0.1500 |
| 0.20 | 0.1937 | 0.8688 | 0.1955 | 0.8677 | 0.2000 |
| 0.25 | 0.2441 | 0.9634 | 0.2451 | 0.9614 | 0.2500 |

These results show that even for the challenging ImageNet, with both the VGG and RESNET architectures, our selective classifiers are extremely effective, and with appropriate coverage compromise, our classifier easily surpasses the best known results for ImageNet. Not surprisingly, RESNET, which is known to achieve better results than VGG on this set, preserves its relative advantage relative to VGG through all $r^*$ values.

## 6 Concluding Remarks

We presented an algorithm for learning a selective classifier whose risk can be fully controlled and guaranteed with high confidence. Our empirical study validated this algorithm on challenging image classification datasets, and showed that guaranteed risk-control is achievable. Our methods can be immediately used by deep learning practitioners, helping them in coping with mission-critical tasks.

We believe that our work is only the first significant step in this direction, and many research questions are left open. The starting point in our approach is a trained neural classifier $f$ (supposedly trained to optimize risk under full coverage). While the rejection mechanisms we considered were extremely effective, it might be possible to identify superior mechanisms for a given classifier $f$. We believe, however, that the most challenging open question would be to simultaneously train both the classifier $f$ and the selection function $g$ to optimize coverage for a given risk level. Selective classification is intimately related to active learning in the context of linear classifiers [6, 11]. It would be very interesting to explore this potential relationship in the context of (deep) neural classification. In this paper we only studied selective classification under the 0/1 loss. It would be of great importance

Table 6: SGR results for Image-Net dataset using RESNET50 top-5 for $\delta = 0.001$

| Desired risk ($r^*$) | Train risk | Train coverage | Test risk | Test coverage | Risk bound($b^*$) |
|---|---|---|---|---|---|
| 0.01 | 0.0080 | 0.3796 | 0.0085 | 0.3807 | 0.0099 |
| 0.02 | 0.0181 | 0.5938 | 0.0189 | 0.5935 | 0.0200 |
| 0.03 | 0.0281 | 0.7122 | 0.0273 | 0.7096 | 0.0300 |
| 0.04 | 0.0381 | 0.8180 | 0.0358 | 0.8158 | 0.0400 |
| 0.05 | 0.0481 | 0.8856 | 0.0464 | 0.8846 | 0.0500 |
| 0.06 | 0.0581 | 0.9256 | 0.0552 | 0.9231 | 0.0600 |
| 0.07 | 0.0663 | 0.9508 | 0.0629 | 0.9484 | 0.0700 |

to extend our techniques to other loss functions and specifically to regression, and to fully control false-positive and false-negative rates.

This work has many applications. In general, any classification task where a controlled risk is critical would benefit by using our methods. An obvious example is that of medical applications where utmost precision is required and rejections should be handled by human experts. In such applications the existence of performance guarantees, as we propose here, is essential. Financial investment applications are also obvious, where there are great many opportunities from which one should cherry-pick the most certain ones. A more futuristic application is that of robotic sales representatives, where it could extremely harmful if the bot would try to answer questions it does not fully understand.

## Acknowledgments

This research was supported by The Israel Science Foundation (grant No. 1890/14)

## Footnotes

[1]Whenever the triplet $S_m$, $\delta$ and $r^*$ is infeasible, the algorithm will return a vacuous solution with zero coverage.

[2]While Theorem 3.2 always holds, we note that if $\kappa_f$ is severely skewed (far from ideal), the bound of the resulting selective classifier can be far from the target risk.

[3]With this small $\delta$, and small number of reported experiments (6-7 lines in each table) we did not perform a Bonferroni correction (which can be easily added).

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
