[Reviews · NeurIPS 2017]

Reviewer 1



The paper proposes a practical scheme of adding selective classification capabilities to an existing neural network. The method consists of: 1. Choosing a score function that captures how confident the network is in its prediction, analysed are MC-dropout scores for networks trained with dropout and the maximum softmax score for networks with a softmax output with the second performing empirically better. 2. Defining the desired confidence level and error rate. 3. Running a binomial search to establish a score threshold such that with the desired confidence level the classifier will have an error rate smaller than the specified one on the samples it chooses to classify. The procedure uses an existing bound on the true error rate of a classifier based on a small sample estimate (Lemma 3.1) and uses binomial search with a Bonferroni correction on the confidence level (Algorithm 1) to find the score threshold. Experimental results validate the approach and show good agreement between the algorithm inputs (desired error rate) and observed empirical error rates on a test set. The strong points of the paper are the practical nature of it (with the softmax response score function the procedure can be readily applied to any pretrained neural network) and the ease of specifying the algorithm’s desired confidence level and error rate (which is modeled after ref [5]). While the paper builds on well known concepts, the careful verification of the concepts adds a lot of value. The paper lacks simple baselines, that could showcase the importance of using the binomial search and the bound on the classifier’s error rate. In particular, I would like to know what happens if one chooses the score threshold as the lowest value for which the error rate on a given tuning set is lower than e specified value- would the results be much more different than using the bound from Lemma 3.1? Knowing this baseline would greatly motivate the advanced techniques used in the paper (and would raise my score of this paper). Nitpicks: the Algorithm 1 uses an uninitialized variable r*

Reviewer 2



Selective classification is the problem of simultaneously choosing which data examples to classify, and subsequently classifying them. Put another way, it’s about giving a classifier the ability to ignore certain data if it’s not confident in its prediction. Previous approaches have focused on assigning a small cost for abstaining. This paper proposes a post-hoc strategy where, if a classifier’s confidence can be accurately gauged, then this confidence is thresholded such that the classifier obtains a guaranteed error rate with high probability. The main novelty with this paper is the proposed SGR algorithm and associated theory. This relies on an ideal confidence function, which is not available in practice, so two methods, SR and MC-dropout are tested. The results are promising, obtaining a low test error with a reasonably high coverage. Getting into specifics: it’s not obvious how you solve Equation (4). I’m assuming it’s a simple line search in 1D, but it would be helpful to be explicit about this. Also, what is the complexity of this whole procedure? It looks like it’s mlog(m)? It’s interesting that mc-dropout performed worse on Imagenet, do you have any intuition as to why this might be the case? It may be helpful to visualize how the confidence functions differ for a given model. I suppose one can easily test both and take the one that works better in practice. As far as I know, there is no explicit validation set for CIFAR-10 and CIFAR-100. They each have 50,000 training points with a separate 10,000-point test-set. Did you split up the test-set into 5,000 points? Or did you use the last batch of the training set for Sm? I think a more proper way to evaluate this would be to use some portion of the last batch of the training sets as validation, and evaluate on the full test set. It would be helpful for you to mention what you did for Imagenet as well; it looks like you split the validation set up into two halves and tested on one half? Why not use the full test set, which I think has 100,000 images? There’s a typo in section 5.3 (mageNet). One point of weakness in the empirical results is that you do not compare with any other approaches, such as those based on assigning a small cost for abstaining. This cost could be tuned to get a desired coverage, or error rate. It’s not clear that a post-hoc approach is obviously better than this approach, although perhaps it is less expensive overall. Overall I like this paper, I think it’s a nice idea that is quite practical, and opens a number of interesting directions for future research.

Reviewer 3



The paper addresses the problem of constructing a classifier with the reject option that has a desired classification risk and, at the same time, minimizes the probability the "reject option". The authors consider the case when the classifiers and an associate confidence function are both known and the task is to determine a threshold on the confidence that determines whether the classifier prediction is used or rejected. The authors propose an algorithm finding the threshold and they provide a statistical guarantees for the method. Comments: - The authors should provide an exact definition of the task that they attempt to solve by their algorithm. The definition on line 86-88 describes rather the ultimate goal while the algorithm proposed in the paper solves a simpler problem: given $(f,\kappa)$ find a threshold $\theta$ defining $g$ in equation (3) such that (2) holds and the coverage is maximal. - It seems that for a certain values of the input arguments (\delta,r^*,S_m,...) the Algorithm 1 will always return a trivial solution. By trivial solution I mean that the condition on line 10 of the Algorithm 1 is never satisfied and thus all examples will be at the end in the "reject region". It seems to me that for $\hat{r}=0$ (zero trn error) the bound B^* solving equation (4) can be determined analytically as $B^* = 1-(\delta/log_2(m))^{1/m}$. Hence, if we set the desired risk $r^*$ less than the number $B^* = 1-(\delta/log_2(m))^{1/m}$ then the Algorithm 1 will always return a trivial solution. For example, if we set the confidence $\delta=0.001$ (as in the experiments) and the number of training examples is $m=500$ then the minimal bound is $B^*=0.0180$ (1.8%). In turn, setting the desired risk $r^* < 0.018$ will always produce a trivial solution whatever data are used. I think this issue needs to be clarified by the authors. - The experiments should contain a comparison to a simple baseline that anyone would try as the first place. Namely, one can find the threshold directly using the empirical risk $\hat{r}_i$ instead of the sophisticated bound B^*. One would assume that the danger of over-fitting is low (especially for 5000 examples used in experiments) taking into account the simple hypothesis space (i.e. "threshold rules"). Without the comparing to baseline it is hard to judge the practical benefits of the proposed method. - I'm missing a discussion of the difficulties connected to solving the numerical problem (4). E.g. which numerical method is suitable and whether there are numerical issues when evaluating the combinatorial coefficient for large m and j. Typos: - line 80: (f,g) - line 116: B^*(\hat{r},\delta,S_m) - line 221: "mageNet"